# Regulation of m^6^A Methylome in Cancer: Mechanisms, Implications, and Therapeutic Strategies

**DOI:** 10.3390/cells13010066

**Published:** 2023-12-28

**Authors:** Poshan Yugal Bhattarai, Garam Kim, Dibikshya Bhandari, Pratikshya Shrestha, Hong Seok Choi

**Affiliations:** College of Pharmacy, Chosun University, Gwangju 61452, Republic of Korea; poshanb@chosun.ac.kr (P.Y.B.); garam1204@korea.kr (G.K.); dibikshya@chosun.ac.kr (D.B.); pratikshyashrestha31@gmail.com (P.S.)

**Keywords:** N^6^-methyladenosine (m^6^A) modification, m^6^A-related proteins, post-translational modification, m^6^A specificity, novel therapeutic targets

## Abstract

Reversible *N*^6^-adenosine methylation of mRNA, referred to as m^6^A modification, has emerged as an important regulator of post-transcriptional RNA processing. Numerous studies have highlighted its crucial role in the pathogenesis of diverse diseases, particularly cancer. Post-translational modifications of m^6^A-related proteins play a fundamental role in regulating the m^6^A methylome, thereby influencing the fate of m^6^A-methylated RNA. A comprehensive understanding of the mechanisms that regulate m^6^A-related proteins and the factors contributing to the specificity of m^6^A deposition has the potential to unveil novel therapeutic strategies for cancer treatment. This review provides an in-depth overview of our current knowledge of post-translational modifications of m^6^A-related proteins, associated signaling pathways, and the mechanisms that drive the specificity of m^6^A modifications. Additionally, we explored the role of m^6^A-dependent mechanisms in the progression of various human cancers. Together, this review summarizes the mechanisms underlying the regulation of the m^6^A methylome to provide insight into its potential as a novel therapeutic strategy for the treatment of cancer.

## 1. Introduction

Ribonucleotides in RNA molecules undergo a diverse range of chemical modifications of nitrogenous bases and ribose sugars. The term “epitranscriptome” collectively refers to all the chemical modifications in RNA molecules. To date, more than 170 types of chemical modifications have been reported in RNAs, including messenger RNA (mRNA), ribosomal RNA (rRNA), transfer RNA (tRNA), long noncoding RNA (lncRNA), and small nucleolar RNA [1]. N^6^-methyladenosine (m^6^A) is the most common internal chemical modification of mRNA. The m^6^A modification is catalyzed by a nuclear methyltransferase complex comprising methyltransferase-like (METTL)3, METTL14, and Wilms’ tumor-associated protein (WTAP) [2]. Proteins with a YT521-B homology (YTH) domain, such as YTHDF1, F2, F3, C1, and C2, specifically recognize and bind to m^6^A nucleotides [3]. In contrast, RNA demethylases with an Alk domain, such as ALKBH5 and ALKBH9 (FTO), can remove methyl groups from m^6^A [3]. In this section, we provide a concise introduction to m^6^A-related proteins and a comprehensive overview of their regulatory mechanisms in cancer cells.

### 1.1. Writers

Writers include proteins involved in the N^6^-adenosine methyltransferase reaction. In the methyltransferase complex, METTL3-METTL14 forms a dimer responsible for m^6^A modification [2]. In contrast, WTAP recruits the METTL3-METTL14 dimer to nuclear speckles, the primary sites of m^6^A deposition, located in the cell nucleus [4]. Recent crystallographic studies have shown that METTL3 is the sole catalytic subunit of the methyltransferase complex [5]. In contrast, METTL14 provides allosteric support to recognize the RNA substrate and helps to stabilize the methyl transferase complex (MTC) [6]. Since METTL3 and METTL14 contain a methyltransferase (MTase) domain, either of these purified proteins can catalyze the methyltransferase reaction in vitro [7]. However, the rate of catalysis is significantly elevated for the METTL3-METTL14 dimer complex [7]. 

### 1.2. Readers

The m^6^A-readers include proteins that recognize and bind to the m^6^A nucleotide. Reader proteins are broadly classified as direct or indirect readers [3]. Direct readers include proteins that directly recognize m^6^A nucleotides. The ability to recognize m^6^A is attributed to the YTH domain. Proteins with a YTH domain share significant homology with other RNA-binding domains. The role of YTH domain-containing proteins as m^6^A readers is reinforced by the finding that an m^6^A probe pulls down the YTH protein from cell lysates [8]. The human genome encodes five proteins with YTH domains: YTH-DF1, -DF2, -DF3, -DC1, and -DC2. YTHDF1, -DF2, and -DF3 are primarily located in the cytosol, -DC1 is localized in the nucleus, and -DC2 is ubiquitous [3]. Indirect readers can recognize the structural deformations in mRNA caused by m^6^A. Given that m^6^A-modified nucleotide exhibits a lower ability to base-pair with uracil than with adenosine, the presence of m^6^A bases in mRNA tends to form a linear, unfolded structure around its location [9]. Structural changes in mRNA caused by m^6^A are known as the “m^6^A structural switch” [10]. Indirect m^6^A readers recognize the m^6^A switch and include two well-known proteins: heterogeneous nuclear ribonucleoprotein G (HNRNPG) and A2B1 (HNRPA2B1) [3]. Indirect readers have a lower binding affinity for unmethylated mRNA than for m^6^A-modified mRNA [3].

### 1.3. Erasers

The m^6^A eraser includes proteins involved in the demethylation reaction, which converts m^6^A into A. The presence of m^6^A erasers in the human genome suggests that m^6^A modification is a dynamic event in which cell signaling pathways can be regulated in response to the extracellular environment. m^6^A erasers are characterized by sequence homology with the ALKB family of dioxygenases [11]. These enzymes, present in diverse organisms ranging from bacteria to higher eukaryotes, use molecular oxygen and α-ketoglutarate to catalyze the oxidative demethylation of specific alkylated bases such as the methyl group in both DNA and RNA. ALKBH5 is a prominent m^6^A eraser protein in mammals [12]. In addition, fat mass and obesity-associated protein (FTO), also known as ALKBH9, has been recognized as m^6^A demethylase. Interestingly, initial studies linked FTO to obesity and body mass regulation. However, subsequent research unveiled its broader role in the regulation of RNA processing. FTO is implicated in the regulation of mRNA splicing, stability, and translation. Recent studies indicate that FTO demonstrates a poor affinity toward m^6^A; instead, it is a strong eraser of the *N*^6^,2′-O-dimethyladenosine (m^6^A_m_) modification [13]. 

## 2. Post-Translational Modification of m^6^A-Related Proteins

A profound understanding of m^6^A-related proteins has prompted inquiries into their regulation in both physiology and diseases. Herein, we summarize the key post-translational modifications (PTMs) of m^6^A-related proteins verified with biochemical experiments and their influence on the regulation of the m^6^A methylome, particularly in cancer cells. Additionally, we discuss the significance of these PTMs from a therapeutic perspective. The major PTMs found in m^6^A proteins and the signaling pathways that govern them are illustrated in Figure 1.

### 2.1. Phosphorylation

Protein phosphorylation is the most common and arguably the most important PTM that regulates cellular processes such as signal transduction, protein synthesis, cell division, and apoptosis [14]. Phosphorylation is mediated by a cascade of protein kinases that are crucial regulators of m^6^A-related proteins. The phosphorylation of METTL3 and WTAP at specific serine sites is driven by the extracellular signal-regulated kinases (ERK) pathway. ERK phosphorylates METTL3 at S43, S50, and S525, while it phosphorylates WTAP at S306 and S341 [15]. This phosphorylation, along with the subsequent deubiquitination by USP5, stabilizes the m^6^A methyltransferase complex. Reduced METTL3/WTAP phosphorylation increases the stability of m^6^A-labeled pluripotent factor transcripts, such as Nanong, and maintains pluripotency in mouse embryonic stem cells [15]. The same phosphorylation pattern, which promotes tumorigenesis, has been observed in breast and melanoma cancer cells [15]. The ERK pathway is often activated by extracellular growth factors such as epidermal growth factor (EGF). Activation of the epidermal growth factor receptor (EGFR) by EGF triggers the phosphorylation of METTL3, which subsequently catalyzes the methylation of small nuclear 7SK mRNA. This methylation event enhances the affinity of 7SK for heterogeneous nuclear ribonucleoproteins, resulting in the dissociation of the HEXIM1/P-TEFb complex and the promotion of transcriptional elongation. This process underscores the intricate role of the MEK/ERK pathway in the regulation of METTL3 function [16].

The S43 site in METTL3 is also phosphorylated by the ataxia telangiectasia mutation (ATM) in response to double-strand breaks (DSBs). The ATM kinase triggers the activation of METTL3, which is subsequently guided to DNA damage sites, where it introduces m^6^A modifications to adenosine within DNA damage-associated RNAs. This modification recruits the m^6^A reader protein, YTHDC1, providing protective functions [17]. The function of METTL3 in DSB repair has provided insights into the role of METTL3 in the radioresistance of cancer cells. The stability of METTL3 phosphorylated at S525 is regulated by the prolyl cis/trans isomerase PIN1. The peptidyl-prolyl isomerase (PPIase) domain of PIN1 interacts with METTL3 and promotes its stability by preventing proteasomal and lysosomal degradation in breast cancer cells [18]. In addition to these phosphorylation events, proteomic analyses have revealed various phosphorylation sites in METTL3 and METTL14. Among these, the phosphorylation of S399 at the C-terminus of METTL14 is intriguing. This modification establishes a salt bridge with R471 in METTL3, suggesting a significant regulatory role in modulating the methyltransferase activity of the MTC complex [6,19].

Proteomic studies have revealed that phosphorylation can control the expression and aggregation of YTHDF proteins [20]. In glioblastomas, it has recently been revealed that DF2 undergoes phosphorylation at serine 39 (S39) and threonine 381 (T381) by ERK1. These phosphorylation events enhance the half-life of DF2 two-fold and significantly boost the invasiveness and proliferation of glioblastoma cells [21].

Casein kinase II (CKII)-mediated phosphorylation of FTO at threonine 150 (T150) plays an important role in nucleocytoplasmic shuttling [22]. Nuclear FTO demethylates CCND1 mRNA, thereby increasing its stability during the G1 phase of the cell cycle. Nuclear localization of FTO is inhibited by CKII-mediated phosphorylation [22].

Reactive Oxygen Species (ROS) play a significant role in increasing global m^6^A levels by regulating ALKBH5 [23]. This rapid and effective upregulation of m^6^A affects thousands of genes, particularly those involved in DNA damage repair. Mechanistically, ROS stimulate ALKBH5 SUMOylation via ERK/JNK signaling-mediated phosphorylation at serine residues S87 and S325, which, in turn, inhibits the m^6^A demethylase activity of ALKBH5 by impeding substrate accessibility. The ROS-triggered ERK/JNK/ALKBH5 pathway is also active in hematopoietic stem/progenitor cells (HSPCs) in vivo in mice, highlighting its physiological significance in safeguarding genomic stability within HSPCs. This cited study revealed a molecular mechanism involving ALKBH5 phosphorylation and increased mRNA m^6^A levels that preserve cellular genomic integrity in response to ROS [23].

### 2.2. Methylation

The methylation of Lys and Arg amino acids on non-histone proteins is a common post-translational modification that regulates signal transduction via various pathways and influences cellular functions, such as chromatin remodeling, gene transcription, and DNA repair [24]. Notably, protein and RNA methylation share the same substrate as the source of the methyl group, that is, S-adenosine methionine (SAM), implying that a detailed understanding of the relationship between these two modifications may offer a more powerful therapeutic option.

Arginine methylation of METTL14 at arginine methyltransferases (PRMT) by arginine 255 (R255) enhances the stability of the interaction between the m^6^A methyltransferase complex and its RNA substrate [25]. This in turn boosts global m^6^A modifications and supports the differentiation of mouse embryonic stem cells (mESCs) into the endoderm. These findings highlight the intricate interplay between protein and RNA methylation in the regulation of gene expression [25]. Consequently, the suppression of PRMT with MS023 inhibits cancer cell proliferation induced by METTL14 overexpression [26]. In addition, PRMT1-induced methylation of WTAP promotes m6A methyltransferase function in multiple myeloma [27].

### 2.3. Acetylation

Protein acetylation uses acetyl-CoA, a component of the cellular metabolic pathway, as a source of an acetyl group [28]. The incorporation of metabolites into the protein structure allows cells to incorporate metabolic cues into intricate cellular decision-making processes such as protein acetylation. Therefore, studies on acetylation of m^6^A-related proteins offer valuable insights into the mechanisms by which cellular metabolism affects the global m^6^A methylome.

METTL3 acetylation regulates its localization and profoundly affects metastatic spread. IL-6, whose mRNA transcript undergoes METTL3-mediated m^6^A modification, promotes METTL3 deacetylation via the nicotinamide adenine dinucleotide (NAD)-dependent histone deacetylase silent information regulator (SIRT1) [29]. Deacetylation of METTL3 promotes its nuclear translocation and consequently elevates global m^6^A levels. This deacetylation-driven shift in METTL3 to the nucleus can be counteracted by inhibiting SIRT1, an effect further potentiated with aspirin treatment, which ultimately impairs lung metastasis. Similarly, acetyl-CoA acetyltransferase 1 (ACAT1) induces METTL3 acetylation and promotes its stability by inhibiting ubiquitin-mediated proteasomal degradation [30]. METTL3 upregulation mediated by ACAT1 is associated with increased migration and invasion of triple-negative breast cancer (TNBC) cells [30].

The catalytic function of m^6^A demethylase ALKBH5 is controlled by acetylation at lysine 235 (K235), which is catalyzed by lysine acetyltransferase 8 and reversed by histone deacetylase 7 [31]. The acetylation of K235 enhances the ability of ALKBH5 to recognize m^6^A modifications in mRNA, and this recognition is further enhanced by the RNA-binding protein PSCP1, which acts as a regulatory subunit of ALKBH5. PSCP1 preferentially interacts with K235-acetylated ALKBH5, leading to the recruitment and improved recognition of m^6^A mRNA, ultimately promoting m^6^A removal. Signals that stimulate cell growth, such as serum, promote the acetylation of K235 in ALKBH5. Notably, K235 acetylation in ALKBH5 is upregulated in cancer and contributes to tumor development. These findings underscore the significance of K235 acetylation in orchestrating the m^6^A demethylation activity of ALKBH5 with the assistance of the regulatory subunit PSCP1, highlighting the crucial role of K235 acetylation in the m^6^A demethylase function and oncogenic activities of ALKBH5 [31].

### 2.4. SUMOylation

SUMOylation is a process that entails the covalent bonding of a protein from the small ubiquitin-like modifier (SUMO) family to lysine residues within target proteins. This occurs through an enzymatic cascade that is similar to the ubiquitination pathway but with distinct characteristics [31]. SUMO1 predominantly modifies METTL3 at lysine residues K177, K211, K212, and K215, and these modifications can be diminished through the action of the SUMO1-specific protease, sentrin/SUMO-specific protease 1 (SENP1) [32]. The SUMOylation of METTL3 does not impact its stability, localization, or interaction with METTL14 and WTAP. However, it notably suppresses its m^6^A methyltransferase activity, leading to a reduction in m^6^A levels within mRNAs. The modification of m^6^A in mRNA molecules, induced by METTL3 SUMOylation, has direct implications for alterations in gene expression profiles. These changes, in turn, can influence processes such as soft-agar colony formation and xenograft tumor growth in H1299 cells [32].

YTHDF2 is SUMOylated both in vivo and in vitro at the major site K571 by small ubiquitin-related modifier 1 (SUMO1), whereas sentrin-specific protease 1 (SENP1) removes the SUMO modification. Additionally, SUMOylation is induced by hypoxia and reduced by oxidative stress and chemical inhibitors [33]. SUMOylation of YTHDF2 has minimal impacts on its ubiquitination and localization. However, the binding affinity for m^6^A-modified mRNAs was significantly increased by SUMOylation. Utilizing RNA immunoprecipitation (RIP), the authors of one study demonstrated that substituting SUMOylated lysine residues with SUMO modification-deficient arginine significantly reduced the interaction between YTHDF2 and PLAC2, a non-coding RNA well-known for binding with YTHDF2. Furthermore, increased binding with YTHDF2 induced the degradation of RNA. The profound changes in transcriptome induced by the YTHDF2 SUMOylation accounted for cancer progression in adenocarcinoma [33].

### 2.5. O-GlcNAcylation

O-GlcNAcylation is a PTM that responds to nutrient availability and stress. This includes the addition of O-linked N-acetylglucosamine groups to serine and threonine residues of proteins. O-GlcNAc modifies the m^6^A mRNA reader YTHDF1 and fine-tunes its nuclear translocation [34]. O-GlcNAc transferase (OGT) binds to YTHDF1 and modifies Ser196/Ser197/Ser198 sites. Moreover, O-GlcNAcylation augments the cytosolic localization of YTHDF1 by strengthening its interaction with chromosomal maintenance 1 (Crm1), also recognized as exportin 1. This enhancement results in the upregulation of translation efficiency for specific downstream targets, including c-Myc, in colon cancer cells.

## 3. Transcriptional Activation of m^6^A-Related Genes

Compared with the PTMs of m^6^A genes, the transcriptional mechanisms that regulate the expression of these genes remain poorly understood. This suggests that the focus should shift toward the investigation of these mechanisms. In particular, gaining a deeper understanding of how the activation of specific cell signaling pathways triggers the transcription of specific m^6^A-related genes holds great promise for the development of novel therapeutic strategies for cancer treatment. Additionally, these mechanisms shed light on the larger picture of how the m^6^A methylome is shaped throughout the various stages of carcinogenesis, ranging from neoplastic transformation to drug resistance.

In gastric cancer (GC) cells, the activation of METTL3 transcription is induced by the promotion of the P300-mediated H3K27 acetylation of its promoter [35]. This, in turn, stimulates an m^6^A modification on hepatoma-derived growth factor (HDGF) mRNA. The m^6^A site on HDGF mRNA is subsequently recognized and binds to the m^6^A reader, insulin-like growth factor 2 mRNA-binding protein 3 (IGF2BP3), resulting in enhanced HDGF mRNA stability. Tumor angiogenesis is promoted by the secretion of HDGF, whereas the activation of GLUT4 and ENO2 expression by nuclear HDGF leads to an increase in glycolysis in GC cells. This increase in glycolysis is associated with subsequent tumor growth and liver metastasis [35].

Cigarette smoke condensation (CSC) induces the hypomethylation of the METTL3 promoter, resulting in elevated expression of METTL3 in pancreatic duct epithelial cells. Following this, the oncogenic primary microRNA-25 (miR-25) undergoes excessive maturation due to cigarette smoke condensate (CSC), facilitated by increased m6A modification mediated by nuclear factor-kappa B-associated protein (NKAP). The mature forms, miR-25 and miR-25-3p, act to suppress PH domain leucine-rich repeat protein phosphatase 2 (PHLPP2), consequently activating the oncogenic AKT-p70S6K signaling pathway and inducing malignant phenotypes in pancreatic cancer cells [36].

The master regulator of the tumorigenesis transcription factor, myelocytomatosis (MYC), binds to the promoter of the m^6^A reader, IGF2BP3, and activates transcription. Moreover, IGF2BP3 promotes the stability of m^6^A-modified KPNA2, leading to cell proliferation and metastasis in nasopharyngeal carcinoma cells [37].

These studies, although limited in number, have demonstrated that the expression of m^6^A-related genes may be influenced by the extracellular environment through transcriptional mechanisms. In particular, exploring how epigenetic modifications affect the expression of m^6^A genes and shape the epitranscriptome is an intriguing topic for further research. Such investigations would illustrate the dynamic interplay between epigenetics and the epitranscriptome in cancer cells.

## 4. Regulation of m^6^A Specificity

Despite a clear understanding of m^6^A methylation machinery, the mechanisms governing the specificity of m^6^A deposition remain elusive. m^6^A-IP-Seq experiments have revealed that m^6^A in human cells is predominantly enriched in the 3′-untreanslated (3′-UTR) region to regulate the translation efficiency and stability of mRNA [38]. In addition, m^6^A is highly enriched in mRNA containing long exons [8]. Furthermore, the m^6^A modification of specific mRNA is changeable depending on extracellular signals, nutrient availability, developmental stage, and response to chemotherapy in cancer cells [39]. The characteristic distribution pattern of m^6^A in the transcriptome raises questions about the mechanisms that regulate its specificity. To date, three potential mechanisms have been reported that underscore the roles of transcription factors, epigenetic modifications, and exon architecture in the pattern of m^6^A methylation, as summarized in Figure 2. These findings collectively indicate a close association between the transcriptional process and m^6^A modification. Specifically, transcription factors and co-activators have been identified as key players in recruiting the m^6^A methyltransferase complex to specific gene promoters, thereby promoting co-transcriptional m^6^A modification of the mRNA transcribed from the respective promoter. While the study of transcription factors has traditionally focused on their role in transcriptional regulation, the presented data highlight an additional role for these factors in post-transcriptional RNA processing. Consequently, their involvement in facilitating m^6^A modification suggests a broader impact on translation efficiency, expanding our understanding of the multifaceted functions of transcription factors beyond their well-established transcriptional regulatory roles. Moreover, epigenetic modifications, such as histone methylation, play a pivotal role as markers in recruiting the methyltransferase complex. Recent studies have highlighted the significant contribution of exon architecture in defining the distinctive pattern of m^6^A modification observed in mRNA. This exon-centric model precisely elucidates the reasons behind the enrichment of m^6^A modifications in the last exon or within long internal exons. However, our understanding of the upstream signaling pathways that influence the distribution of m^6^A marks across the transcriptome remains incomplete. The specific factors contributing to m^6^A specificity are detailed below, shedding light on the intricate network of regulatory elements that govern the precise localization of m^6^A modifications on RNA transcripts.

### 4.1. Transcription Factors and Co-Activators

The first evidence of specificity comes from a study on the role of METTL3 in acute myeloid leukemia (AML). METTL3, separate from METTL14, binds to chromatin and is positioned at the transcriptional start sites of actively expressed genes [40]. The majority of these genes harbor the CAATT-box-binding protein CCAAT enhancer binding protein zeta (CEBPZ) at their transcriptional start sites, and this presence is essential for recruiting METTL3 to chromatin. METTL3, when bound to the promoter, instigates m^6^A modifications within the coding region of the corresponding mRNA transcript. This action, in turn, promotes enhanced translation by alleviating ribosomal stalling. The identified mechanism establishes METTL3 as a pivotal protein for sustaining the leukemic state, emphasizing its therapeutic potential in the context of acute myeloid leukemia.

A study of the protein interactome of SMAD2/3 transcription factors, activated by the transforming growth factor beta (TGFβ) signaling pathway, in human pluripotent stem cells revealed a functional interaction with the m^6^A methyltransferase complex [41]. SMAD2/3 promotes the binding of the complex to a subset of transcripts involved in early cell fate decisions, such as NANOG. The m^6^A modification of these transcripts induces mRNA degradation to enable exit from pluripotency. Although these findings have been reported in stem cells, they may have far-reaching implications for cancer [41].

### 4.2. Epigenetic Modification

Trimethylation of Lys36 on histone H3 (H3K36me3), an indicator of transcriptional elongation, plays a pivotal role in guiding the global deposition of m^6^A [42]. The presence of H3K36me3 peaks is associated with the enrichment of m^6^A modifications, and depletion of the overall levels of H3K36me3 leads to a global reduction in m^6^A. METTL14 directly recognizes and binds H3K36me3. This interaction facilitates the binding of the m^6^A methyltransferase complex to adjacent RNA polymerase II, allowing the complex to co-transcriptionally deposit m^6^A on actively transcribed nascent RNAs. In mouse embryonic stem cells, a decrease in H3K36me3 expression significantly lowers m^6^A levels across the transcriptome, particularly in pluripotency transcripts, consequently enhancing cell stemness. These findings underscore the crucial role of epigenetic signatures, such as H3K36me3 and METTL14, in orchestrating the specific and dynamic deposition of m^6^A in mRNA.

### 4.3. Exon Architecture

Exon junction complexes (EJCs) have been recognized as inhibitors of m^6^A methylation, safeguarding RNA near exon junctions within coding sequences from this modification [43,44,45]. They also control mRNA stability by inhibiting m^6^A addition. The ability of the EJC to prevent m^6^A methylation contributes to several key features of mRNA m^6^A specificity. The protective influence of EJCs extends over the local region, effectively preventing m^6^A addition to average-length internal exons but not to lengthy internal and terminal exons. This mechanism also accounts for a preferential enrichment in m^6^A in the 3’ UTR and extended internal exons. Additionally, sites where EJCs inhibit methylation coincide with locations where EJCs suppress splice sites, suggesting that the exon structure broadly influences the accessibility of mRNA to regulatory complexes [43].

Similarly, various studies have revealed the role of the EJC in molding the m^6^A modification landscape. Specificity is achieved by inhibiting METTL3-mediated m^6^A modification in the vicinity of exon junctions within coding sequences (CDS). When EIF4A3, a core component of the EJC, is depleted, it leads to heightened METTL3 binding and increased m^6^A modification in short internal exons and sites near exon–exon junctions within mRNA. These reports shed light on the mechanisms underlying the establishment of distinct m^6^A mRNA modification patterns and underscore the influence of the EJC on shaping the m^6^A epitranscriptome [45].

## 5. Dysregulation of m^6^A in Cancer and Chemoresistance

Because of the numerous post-translational modifications (PTMs) discussed earlier, the expression levels of m^6^A-related proteins and their functions are frequently disrupted in cancers. Additionally, the abnormal expression of specific m^6^A-related genes, such as METTL3 and ALKBH5, contributes to the development of resistance to cytotoxic chemotherapy, targeted drugs, and immunotherapy. Moreover, METTL3, an m^6^A methyltransferase, plays a positive role in double-strand break repair, further promoting resistance to radiotherapy. Target mRNAs subject to m^6^A methylation in cancer appear to be critical for cancer cell proliferation and survival, encompassing genes related to autophagy, the cell cycle, and proliferation, among others (summarized in Table 1). Despite the abundance of information on m^6^A-modified mRNA, the mechanisms underlying the specificity of m^6^A modifications remain unclear. While the exon architecture and epigenetic mechanisms discussed earlier explain the patterns of m^6^A methylation across the transcriptome, they do not address how cell signaling pathways regulate the m^6^A modification of specific mRNAs. Recruitment of m^6^A methyltransferase subunits such as METTL3 by specific transcription factors may be a possible mechanism governing the specific and dynamic m^6^A modification of mRNA. However, the underlying mechanisms remain poorly understood in cancer. Additionally, a comprehensive understanding of the interactome of m^6^A-releated proteins may reveal novel mechanisms regulating the functions of m^6^A-related genes. A recent study on the protein interactome of m^6^A methyltransferases and demethylases revealed numerous novel binding partners for these proteins; however, the functional characterization of these interactions remains to be studied [46].

## 6. Future Perspective

A better understanding of m^6^A-dependent mechanisms during carcinogenesis has paved the way for the development of small-molecule inhibitors that target m^6^A-related proteins. Inhibitors targeting METTL3 and FTO have been developed, and their inhibitory functions have been evaluated in cell lines and in vivo models (Table 2). However, no compounds have yet been approved for use in clinical trials. STM2547 is a potent first-in-class catalytic inhibitor of METTL3 [56]. Treatment with STM2457 significantly reduced acute myeloid leukemia tumor formation in a mouse model. Furthermore, STC-15, a compound derived from STM2457 with excellent oral bioavailability, has demonstrated significant anticancer efficacy against AML in preclinical models [57] and is currently enrolled in phase I human clinical trials [57] (ID: NCT05584111; clinicaltrials.gov). Table 2 provides a summary of chemical inhibitors targeting m^6^A-related proteins documented in the literature.

## 7. Conclusions

In summary, we presented an overview of our current knowledge regarding the regulation of m^6^A modification in cancer cells. Current insights suggest that the m^6^A methylome in cancer cells is predominantly influenced by the post-translational modification of proteins associated with m^6^A. These modifications either modulate the catalytic activities of m^6^A methyltransferases and demethylases or impact the ability of m^6^A reader proteins to interact with and process m^6^A-modified mRNA. The resulting alterations in the expression and fate of m^6^A-modified mRNA carry significant implications for carcinogenesis. Additionally, the specificity of m^6^A deposition is shaped by factors such as transcription factors, epigenetic modifications, and the inherent architecture of exons. Future investigations may unveil whether post-translational modifications of transcription factors or methyltransferase subunits guide the methyltransferase complex to specific gene promoters. A comprehensive understanding of the molecular mechanisms regulating the m^6^A methylome will not only shed light on the intricacies of gene expression in cancer cells but also reveal potential therapeutic targets for cancer treatment.

## Figures and Tables

**Figure 1 cells-13-00066-f001:**
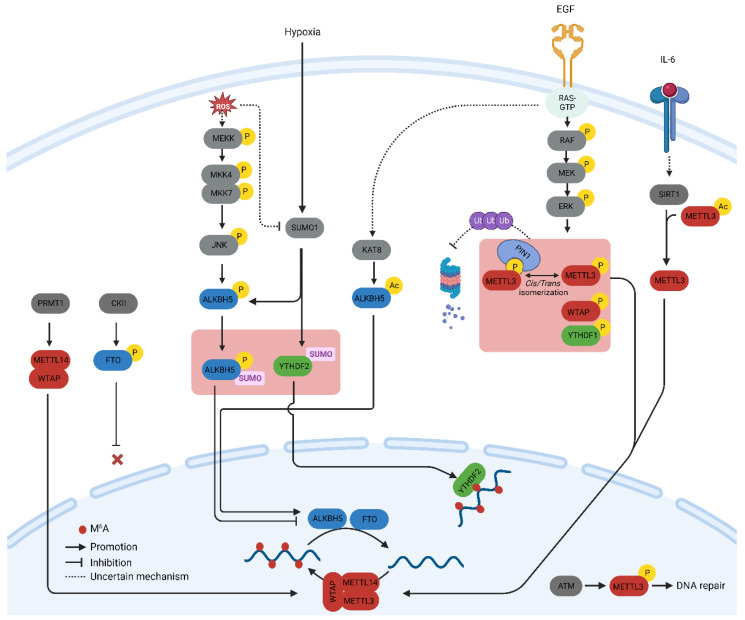
Schematic depiction of signaling pathways orchestrating the post-translational modification of m^6^A-related proteins.

**Figure 2 cells-13-00066-f002:**
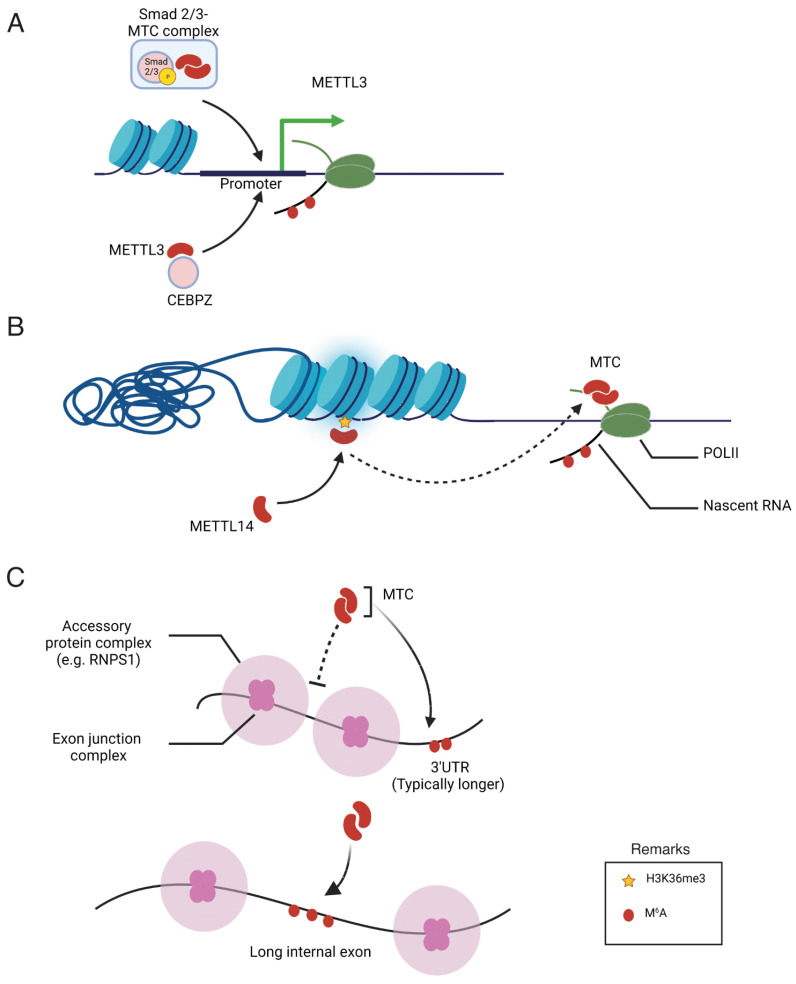
Illustration of proposed mechanisms regulating the specificity of m^6^A deposition. (**A**) Recruitment of the m6A methyltransferase complex (MTC) and METTL3 by Smad2/3 and CEBPZ, respectively, into the promoter of specific genes. (**B**) MTC recruitment to adjacent RNA polymerase II (POLII) is promoted through the interaction of METTL14 with H3K36 trimethylation (H3K36Me3). (**C**) Regulatory effects of exon junction complexes and accessory protein complexes on the m6A modification of specific mRNA.

**Table 1 cells-13-00066-t001:** The role of METTL3 in various human cancers.

Cancer Type	Expression	mRNA Substrate	Biological Function	Phenotype	Ref.
Lung cancer	METTL3, Increased	EGFR, BRD4, MGMT, TIMP1	Translation ↑	Tumorigenesis	[47]
Glioma	METTL3, Increased	SOX2	mRNA stability ↑	Stem-like cell maintenance and radioresistance	[48]
Leukemia	METTL3, Increased	c-My cBCL2 PTEN	Translation ↑	Inhibits differentiation and increases cell growth	[49]
Leukemia	METTL3, Increased	SP1, SP2	Translation ↑	Cell proliferation and differentiation	[40]
Colorectal cancer	METTL3, Increased	SOX2	mRNA stability ↑	Self-renewal, stem cell frequency, and migration	[50]
Ovarian cancer	METTL3, Increased	AXL	Translation ↑	Promotes the proliferation, invasion, and tumor formation	[51]
Melanoma	METTL3, Increased	MMP2	Translation ↑	Promotes invasion	[52]
HCC	METTL3, Increased	SOCS2	mRNA stability ↑	Cell proliferation, migration, and colony formation	[53]
Breast	METTL3, Increased	BCL2 EZH2	mRNA stability ↑ Translation ↑	Promotes proliferation and inhibits apoptosis	[54,55]

The upward arrow indicates the increased biological function.

**Table 2 cells-13-00066-t002:** Chemical inhibitors of m^6^A-related proteins.

Compound	Mechanism of Action	IC_50_	Ref.
STM2457	METTL3 inhibitor	16.9 nM	[56]
Ebselen	YTHDF1/DF2	1.63/1.66 µM	[58]
UZh1a	METTL3 inhibitor	280 nM	[59]
Eltrombopag	Allosteric inhibitor of METTL3-14 complex	4.55 µM	[60]
FB23	FTO inhibitor	60 nM	[61]
FB23-2	FTO inhibitor	2.6 µM	[61]
Dac51	FTO inhibitor	0.4 µM	[62]
Bisantrene	FTO inhibitor	712.8 nM	[63]
Meclofenamate sodium	FTO inhibitor	7 µM	[64]

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
