# Peer review of "Regulation of m6A Methylome in Cancer: Mechanisms, Implications, and Therapeutic Strategies"

_cells, 2023, doi:10.3390/cells13010066_

Round 1

Reviewer 1 Report

Comments and Suggestions for Authors

Dear,

This paper explores various aspects of post-translational modifications (PTM) of mRNAs associated with m6A methylation in the context of carcinogenesis. Different PTMs, including SUMOylation and O-GlcNAcylation, and their impact on proteins related to m6A methylation, such as YTHDF1 and YTHDF2, are presented. Additionally, transcriptional regulations, mechanisms of m6A specificity, and its influence on carcinogenesis and chemoresistance are discussed.

Positive Aspects:

o   The paper covers various research aspects, from PTMs to transcriptional regulations, providing a comprehensive overview of mechanisms related to m6A methylation.

o   Specific examples, such as the activation of METTL3 transcription and the impact of H3K36me3 on m6A levels, contribute to understanding mechanisms in concrete contexts.

o   The discussion of potential therapies through the inhibition of m6A-related proteins, such as STM2547, brings relevance and perspective to the field.

Critiques:

1.     Although mentioned that SUMOylation of YTHDF2 increases binding to m6A-modified mRNAs, additional details about specific mechanisms and consequences of this modification are lacking.

2.     The paper could have a stronger conclusion summarizing key findings and the need for future research.

3.     Some parts of the text, especially in the section on m6A specificity, could be clearer with additional explanations or diagrams.

Conclusion:

The paper provides significant insights into the complexity of m6A methylation regulation and its impact on carcinogenesis. Additional clarifications in some parts, together with a stronger conclusion, could enhance the overall impression of the paper. This research contributes to understanding the role of m6A methylation in carcinogenesis and lays the foundation for further studies in this field.

Best Regards

Reviewer 2 Report

Comments and Suggestions for Authors

Abstract is brief, but provides enough information about the aim of the review, please just add concluding sentence.

1.3 subchapter should be completed with more details and examples

Figs.  are very good illustrations, added value to the manuscript

In general the review presents good insight into the molecular mechanisms related to m6A, summarizing the knowledge about the processes. However the whole elaboration lacks of examples. The reviewer would put more emphasis on the last paragraphs which are presented very briefly.

Written correctly, allowing smooth reading

Actual references list

Comments on the Quality of English Language

The reviewer didn't find serious language faults, correct grammar

Round 2

Reviewer 1 Report

Comments and Suggestions for Authors

Dear,

I want to express sincere gratitude for your dedicated effort. Thank you for accepting suggestions. I want to commend your selfless work and convey the belief that your manuscript has now reached the high standards necessary for publication.

Respectfully

Reviewer 2 Report

Comments and Suggestions for Authors

Thank you very much for following of all my suggestions. The manuscript in current from is more informative. I do not have any further comments.